

# Origin of the X-chromosome influences the development and treatment outcomes of Turner syndrome

Ying Zhang[1], Yongchen Yang[2], Pin Li[1] and Sheng Guo[1]

[1] Department of Endocrinology, Shanghai Children's Hospital, Shanghai Jiaotong University, Shanghai, China
[2] Department of Laboratory Medicine, Shanghai Children's Hospital, Shanghai Jiaotong University, Shanghai, China

## ABSTRACT

Turner syndrome (TS) affects 1/2,500 live-born female infants. In the present study, we attempted to clarify the relationship between genetic factors (especially the X-chromosome origin), clinical features, body/sexual development, and treatment outcomes. We enrolled 39 female infants aged between 3 and 14 years. General demographic and clinical features were documented, and laboratory analysis of blood samples was performed. Subject karyotype was determined by G-banding of 50 peripheral white blood cells, and the parenteral origin of the retained X-chromosome was determined. Next, growth hormone (GH) treatment was prescribed for 12 months, with follow-ups performed as determined. For patient groups separated according to X-chromosome origin, the basal height, bone age, insulin-like growth factor (IGF)-1, and insulin-like growth factor binding protein-3 (IGFBP-3) levels were comparable; however, after the 12-month treatment, significant differences in the height increase and IGF-1 levels were observed. If the X-chromosome (or chromosomes) originated from both parents, the increase in height was less substantial, with lower serum IGF-1 levels. The uterine size, prolactin level, increased weight after treatment, and bone age difference after treatment negatively correlated with the mother's age at the time of birth. The mother's height at the time of birth demonstrated a negative correlation with the basal bone age difference and a positive correlation with the IGF-1 level. In summary, the retained X-chromosome derived from both parents is associated with poorer response to GH therapy. The mother's age and height at the time of birth can strongly impact the patient's body/sexual development and the response to GH treatment. Thus, the mother's age and height at the time of birth and the parental origin of the X-chromosome should be carefully considered before developing a treatment plan for TS.

Corresponding author
Sheng Guo,
guosheng@shchildren.com.cn

## INTRODUCTION

Turner syndrome (TS) is a common chromosomal disorder caused by the loss of all or a significant part of one X-chromosome in a phenotypic female, usually presenting with short stature and premature ovarian failure. This condition affects approximately 1/2,500 live-born female infants. Previous studies have described the cytogenetic characterization

of patients with TS (*Chauhan et al., 2016*; *Burégio-Frota et al., 2010*; *Moka et al., 2013*), with a majority of published reports focusing on karyotype-phenotype associations (*Noordman et al., 2019*; *Cameron-Pimblett et al., 2017*; *Aversa et al., 2015*; *Verver et al., 2014*; *Verver et al., 2011*). For example, a study assessing karyotype features associated with TS-specific hearing loss has revealed that elevated air conduction thresholds can be associated with loss of the p arm of chromosome X (*King et al., 2007*). In contrast, monosomy karyotype (45, X) has been associated with a more severe, lymphatic, and skeletal phenotype of dysmorphic features and cardio-aortic malformations (*Noordman et al., 2018*). The clinical features of TS largely depend on the involved regions of the X-chromosome. These X-structural abnormalities include deletions, duplications, isochromosomes of the long arm, isodicentric chromosomes, complex abnormalities with combined deletions and duplications, inversions, rings, and translocations. Typically, around 40–50% of patients exhibit monosomy (45, X), around 10% of patients are diagnosed with mosaicism (45, X)/(46, XX), approximately 20% of patients have karyotype with isochromosomes, and nearly 20% present other karyotypes (*Noordman et al., 2019*; *King et al., 2007*; *Noordman et al., 2018*; *Ko et al., 2010*; *Alvarez-Nava et al., 2013*). Theoretically, different karyotypes could influence phenotypes during body development, as well as impact treatment outcomes; however, studies investigating these aspects remain limited. Known risk factors associated with development and treatment outcomes predominantly include elevated blood pressure in childhood, mid-parental height (MPH), and genetic factors (*e.g.*, the SHOX deficiency phenotype) (*Nathwani et al., 2000*; *Scalco et al., 2019*; *Jung et al., 2020*; *Child et al., 2015*). Theoretically, some potential predictors are worth investigating, including prolactin, mother's age, and height at the time of birth, and parental origin of the X-chromosome. The parental origin of the X chromosome may impact the disease onset in terms of different aspects (*e.g.*, aortic stiffness) (*Binkert et al., 2010*; *Yeh et al., 2017*; *Zeng et al., 2014*). There have been very few studies investigating the influence of parental origin on treatment results, and the controversy regarding the impact of the retained X-chromosome on clinical features persists. In the present study, we attempted to clarify the relationship between genetic factors (especially the X origin), clinical features, body/sexual development, and treatment outcomes in TS. For the first time, we observed some novel links between the X origin and growth hormone (GH) therapy outcomes, as well as an impact of maternal genetic factors on body/sexual development and GH responses.

## MATERIALS AND METHODS

### Patients

Female children who visited the examination center from January 2019 to April 2020 were selected, and those diagnosed with TS were enrolled. All recruited patients provided signed informed consent. The inclusion criteria for TS children were as follow: (1) females, aged between 3 and 14 years old, (2) diagnosed with TS according to chromosome examination results, (3) who had received GH treatment. TS was diagnosed by lymphocyte chromosomal analysis, in combination with an analysis of clinical features. None of the subjects had received GH therapy or anabolic steroids previously. Potentially relevant

demographic data were recorded, including age, weight, height, father's age at birth, mother's age at birth, father's height at birth, and mother's height at birth. Abnormalities in uterine and ovary size (small/normal/large) were recorded according to previously reported standards (*Cohen et al., 1993*; *Cohen, Tice & Mandel, 1990*; *Orsini et al., 1984*).

## Laboratory analysis

For each subject, whole blood samples from an antecubital vein were collected and immediately centrifuged. The serum was stored or sent to the laboratory within 3 h. Subsequently, the following serum biochemical indicators were measured using the enzymatic method: GH level, GH excitation peak, alanine aminotransferase (ALT), aspartate aminotransferase (AST), total cholesterol, high density lipoprotein (HDL), low density lipoprotein (LDL), insulin, Free T3, Free T4, triiodothyronine, thyroxine, thyroid-stimulating hormone (TSH), insulin-like growth factor (IGF)-1, insulin-like growth factor binding protein-3 (IGFBP-3), luteinizing hormone (LH), follicle-stimulating hormone (FSH), estradiol (E2), prolactin, progesterone, human chorionic gonadotropin (HCG), and sex hormone-binding globulin (SHBG).

## Parental origin of the X-chromosome using X-chromosome short tandem repeats analysis

We performed chromosome karyotype analysis and microsatellite analysis for each patient to jointly infer the parental origin of the X-chromosome. Karyotype analysis is a classic technique for detecting abnormal chromosome numbers, large fragment copy number variation, and chromosome fusion. We used chromosome G-banding technology to analyze 550 bands of chromosomes; microsatellite analysis was then performed for each patient and the parents. The karyotype analysis of chromosomes demonstrates good consistency with the microsatellite analysis. Fluorescence *in situ* hybridization using X- and Y-specific α satellite DNA probes was used to characterize marker-chromosomes and ring-chromosomes. To analyze the influence of different karyotype groups, the following types were defined: (1) Type-1: 45, X (all cells had karyotype 45, X); (2) Type-2: 46, Xdel (X); (3) Type-3: both 46, Xdel (X)(q) or 46, Xdel(X)(q) and 45, X; (4) Type-4: other types. We excluded the karyotype with Y chromosomes. In all samples, the fragmented X-chromosome was present in 20% to 100% of white blood cells. The parental origin of the X-chromosome was determined by detecting nine X-chromosome short tandem repeat (X-STR) loci on the X-chromosome (GATA172D05, DXS10159, DXS6797, HPRTB, DXS10079, DXS6789, DXS9895, DXS10146, and GATA31E08), which are highly polymorphic in the Han population (*Huang et al., 2015*); their positions on the X-chromosome were q23.00, c, q22.30, q26.20, q12.00, q21.33, p22.32, q28.00, and q27.10, respectively. Polymerase chain reaction (PCR) amplification of 9 X-STR loci was performed, followed by a comparison of the patient's result *versus* the mother and the father. The result of the X origin was documented for further analysis.

**Table 1 Demographic information of enrolled patients and general chromosome genotypes.**

| Index | | Value or number | % |
|---|---|---|---|
| Age (month) | | 90.90 ± 28.99 (44–154) | |
| Weight (kg) | | 20.69 ± 5.10 (12.60–36.50) | |
| Height (cm) | | 108.72 ± 11.73 (88.9–131.4) | |
| Father's age when born | | 28.59 ± 1.83 (25–35) | |
| Mother's age when born | | 26.74 ± 1.80 (23–31) | |
| Father's height (cm) | | 169.41 ± 3.73 (160–177) | |
| Mother's height (cm) | | 157.59 ± 3.07 (152–165) | |
| Karyotyping | Type 1 | 15 | 38.5% |
| | Type 2 | 4 | 10.3% |
| | Type 3 | 17 | 43.6% |
| | Type 4 | 3 | 7.7% |
| Source of variation | Father | 6 | 15.4% |
| | Mother | 16 | 41.0% |
| | Both | 17 | 43.6% |

## GH treatment and follow-up

All patients received GH therapy for 12 months (0.15 IU/d/kg), and the height increase, bone age, and IGF-1 and IGFBP3 levels were examined following treatment. Patients lost during follow-up were excluded.

### Statistical analysis

All categorical data were expressed as frequencies and percentages; continuous data were expressed as the mean ± standard deviation (SD). SPSS software 25.0 (IBM Corp., Armonk, NY, USA) was used for statistical analysis. The Chi-Square test was used to compare frequencies in different groups, and one-way ANOVA was used to compare continuous data. A $p$-value $< 0.05$ was set as the significance level.

### Ethics statement

This study was approved by Ethics Review Committee, Children's Hospital of Shanghai/ Shanghai Children's Hospital, Shanghai Jiao Tong University (Approval No: 2017R027-E04), and we strictly followed the standard of ethics established by the committee.

## RESULTS

### Clinical characteristics of included children

A total of 39 patients were enrolled, whose demographic information is presented in Table 1. The average age was 90.90 months (44–154), the average weight was 20.69 kg (12.60–36.50), and the average was height was 108.72 cm (88.9–131.4). The mean age of the patients' fathers at birth was 28.59 years (25–35), and that of mothers was 26.74 years (23–31). The average height of the patients' fathers was 169.41 cm (160–177), and that of mothers was 157.59 cm (152–165). Blood biochemical indicators for enrolled patients are

 

**Table 2 Blood biochemical indicators of enrolled patients.**

| Index | N | Min | Max | Mean | SD |
|---|---|---|---|---|---|
| GH excitation peak | 38 | 3.89 | 19.60 | 9.73 | 3.37 |
| GH (IU/d/kg) | 39 | 0.15 | 0.2 | 0.154 | 0.013 |
| ALT (U/L) | 39 | 6 | 43 | 15.72 | 6.82 |
| AST (U/L) | 39 | 19 | 61 | 32.44 | 7.98 |
| Total cholesterol (mmol/l) | 38 | 2.98 | 6.11 | 4.97 | 0.86 |
| HDL (mmol/l) | 38 | 1.16 | 2.28 | 1.68 | 0.25 |
| LDL (mmol/l) | 35 | 1.40 | 3.64 | 2.64 | 0.50 |
| Insulin (pmol/l) | 39 | 3.02 | 112.00 | 39.60 | 27.67 |
| Free T3 (pmol/l) | 39 | 4.46 | 9.10 | 6.16 | 0.92 |
| Free T4 (pmol/l) | 39 | 11.06 | 26.77 | 13.67 | 2.58 |
| Triiodothyronine (nmol/l) | 39 | 1.33 | 2.62 | 2.02 | 0.27 |
| Thyroxine (nmol/l) | 38 | 87.88 | 140.64 | 112.27 | 11.31 |
| TSH (uIU/ml) | 39 | 0.01 | 17.98 | 3.51 | 2.64 |
| IGF-1 (ng/ml) | 38 | 36.7 | 375.0 | 144.09 | 71.90 |
| IGFBP3 (ug/ml) | 39 | 1.76 | 5.11 | 3.55 | 0.89 |
| LH (IU/L) | 39 | 0.10 | 25.20 | 0.83 | 4.01 |
| FSH (IU/L) | 39 | 0.62 | 133.00 | 6.86 | 20.94 |
| E2 (pmol/l) | 39 | 73.0 | 277.0 | 84.26 | 33.06 |
| Prolactin (mIU) | 39 | 110.00 | 772.74 | 357.89 | 132.71 |
| Progesterone (nmol/l) | 39 | 0.19 | 2.01 | 0.72 | 0.42 |
| HCG (mIU/ML) | 39 | 0.20 | 1.38 | 0.52 | 0.18 |
| SHBG (nmol/l) | 39 | 28.00 | 109.00 | 59.75 | 22.82 |

Note:
GH, growth hormone; HDL, high density lipoprotein; LDL, low density lipoprotein; TSH, thyroid-stimulating hormone; LH Luteinizing hormone; FSH, Follicle-Stimulating Hormone; HCG, human chorionic gonadotropin; SHBG, sex hormone binding globulin; SD, Standard deviation.

presented in Table 2. Overall, patients had relatively low GH, free T4, prolactin, and progesterone levels.

The genotypes of chromosomes were as follows: 38.5% (15 cases) children showed a type-1 chromosome category, 10.3% (4 cases) belonged to type-2, 43.6% (17 cases) were the type-3 chromosome category, and 7.7% (3 cases) belonged to type-4. Based on the parenteral chromosome examination, the X-chromosome(s) origins were classified into three categories: from father, from mother, and from both. Overall, 15.4% (six cases) had X chromosomes from the father, 41.0% (16 cases) had X chromosomes from the mother, and 43.6% (17 cases) had X chromosomes from both.

## Association between X origin and treatment effects

Firstly, we observed the association between the X origin and treatment outcomes. According to the retained X-chromosome origin (father, mother, or both), all patients were divided into three groups. The basal height, bone age, and IGF-1 and IGFBP3 levels were comparable between groups; however, significant differences in the height increase (HI) ($p < 0.001$) and IGF-1 levels ($p < 0.001$) were observed following GH therapy

**Table 3 The impact of X origin on treatment effects.**

| | Father | | Mother | | Both | | F | p |
|---|---|---|---|---|---|---|---|---|
| | **Mean** | **SD** | **Mean** | **SD** | **Mean** | **SD** | | |
| **All samples** | N = 6 | | N = 16 | | N = 17 | | | |
| Height | | | | | | | | |
|     Baseline (cm) | 103.02 | 7.43 | 110.92 | 12.20 | 108.65 | 12.36 | 0.991 | 0.381 |
|     Increase (cm) | 8.00 | 0.00 | 8.24 | 1.53 | 6.25 | 1.26 | 10.693 | 0.000 |
| IGF-1 | | | | | | | | |
|     Baseline (ng/ml) | 127.50 | 49.31 | 157.17 | 90.86 | 137.24 | 57.81 | 0.483 | 0.621 |
|     12-month (ng/ml) | 270.83 | 48.22 | 318.31 | 70.37 | 213.71 | 52.05 | 12.589 | 0.000 |
| **Age matched** | N = 5 | | N = 10 | | N = 12 | | | |
| Height | | | | | | | | |
|     Baseline (cm) | 104.28 | 7.56 | 103.76 | 9.06 | 106.38 | 8.10 | 0.291 | 0.750 |
|     Increase (cm) | 8.00 | 0.00 | 8.31 | 1.24 | 6.56 | 1.31 | 6.744 | 0.005 |
| IGF-1 | | | | | | | | |
|     Baseline (ng/ml) | 137.40 | 48.01 | 104.77 | 45.26 | 127.52 | 35.60 | 1.295 | 0.292 |
|     12-month (ng/ml) | 284.80 | 37.99 | 302.80 | 66.81 | 203.58 | 42.66 | 10.736 | 0.000 |

Note:
SD, Standard deviation.

(Table 3). If the X-chromosome (or chromosomes) was/were derived from both parents, the HI and serum IGF-1 levels were lower than those observed in any parent. Furthermore, the age range of subjects enrolled in this study varied widely. To avoid the impact of age on HI, we used age-matched samples to ensure that only those subjects whose age was within the 90.90 ± 28.99 range of total samples were selected. In this set, the results were highly consistent with the total samples (Table 3). These findings indicated more significant challenges in treating those patients who present chromosome disorders from both parents.

## Maternal genetic factors influencing the development and GH treatment

Theoretically, maternal genetic factors have a greater impact on child development and GH treatment. Among the basal clinical characteristics, uterine size, and prolactin level both negatively correlated with the mother's age at birth of the subject ($p < 0.01$) (Table 4). For prolactin, the relevance was above 0.5 (−0.508), with a high significance ($p = 0.001$). Following GH therapy, the weight increase and bone age difference negatively correlated with the mother's age at birth of the subject ($p < 0.05$). These results may suggest that an elder pregnant woman may bear a TS child with a poor status and more significant treatment challenges. Moreover, the mother's height at the subject's birth may impact development and GH treatment. Accordingly, the basal bone age difference was negatively associated with the mother's height (R = −0.336, $p = 0.036$), while the IGF-1 level after treatment was positively correlated with the mother's height (R = 0.388, $p = 0.015$). However, these relationships have not been observed in paternal genetic factors.

**Table 4 Association between mother's age and height when born on development and GH treatment.**

| Factors | Spearman R | *P* value |
|---|---|---|
| **Association between mother's age when born and following factors** | | |
| Uterine size (small/normal/large) | −0.442 | 0.005 |
| Prolactin (mIU) | −0.508 | 0.001 |
| Weight increase after treatment | −0.351 | 0.028 |
| Bone age difference after treatment | −0.353 | 0.028 |
| **Association between mother's height when born and following factors** | | |
| Bone age difference (baseline) | −0.336 | 0.036 |
| IGF-1 level after treatment | 0.388 | 0.015 |

## DISCUSSION

TS is typically caused by karyotype variations, but it can also be heritable according to some typical case reports (*Ramachandram et al., 2013*; *Periquito et al., 2016*). GH therapy is the most widely used strategy, apart from estrogen and oxandrolone therapies. This treatment can effectively improve the bone mineral density and body height in patients with TS (*Wasniewska et al., 2013*; *Soucek et al., 2011*; *Menke et al., 2010*; *Bannink et al., 2009*; *Spiliotis, 2008*; *Morin et al., 2009*; *Lanes et al., 2019*). In particular, the GH-IGF-IGFBP axis is altered in TS (*Gravholt et al., 2006*); long-term GH treatment can restore IGF-1 and IGFBP-3 levels, which are valuable indicators of treatment outcomes (*Pankowska, Szalecki & Romer, 2007*; *Baş et al., 2012*; *Bautembach-Minkowska et al., 2018*; *Blanco-López et al., 2020*). Notably, IGF-I has been considered a marker of growth response (*Darendeliler et al., 2007*). Moreover, some laboratory indicators are potentially relevant to TS, *e.g.*, prolactin, as implied by previous reports; hence, we included several candidate factors in laboratory analysis (*Amendt, Hesse & Rohde, 1992*; *Salvarci & Zamani, 2018*; *Yeh et al., 2017*). In the present study, we performed a follow-up survey, mainly focusing on changes in height, body weight, IGF-1, IGFBP-3, and bone age.

Generally, the TS karyotype can be classified into four types: monosomy, mosaic, variant, and mosaic with variant (*Wu & Li, 2019*). Monosomy 45, X has been detected in approximately 45–50% of cases, with other patients presenting various chimeras and structural abnormalities (*Cui et al., 2018*). Theoretically, genetic factors can strongly decide the disease state and treatment outcomes. Growth retardation tends to be more severe in patients with XrX, isoXq, and (45, X) karyotypes than in patients with (45, X)/(46, XX) karyotypes or a Y chromosome (*Fiot et al., 2016*). Several studies have assessed the effect of parental origin of the X-chromosome on clinical features, associated complications, and the response to GH therapy. Francisco *Alvarez-Nava et al. (2013)* have shown that parental origin of the retained X-chromosome may influence lipid metabolism in patients with TS, but its effect on growth seems to be minimal. The authors did not detect any impact on phenotypic features, associated anomalies, or the growth response to GH in (45, X) TS individuals. This negative result corroborates the conclusions of *Devernay et al. (2012)* and *Ko et al. (2010)*, although *Kochi et al. (2007)* previously reported a strong influence of genes

located on the maternal X-chromosome, particularly on stature. To date, the present study is among few research endeavors revealing the impact of X-chromosome origin on GH treatment effects. We noted that when a patient with TS presents an X-chromosome (or X chromosomes) from both parents, she is more likely to have a lower HI and IGF-1 level; however, no difference was observed in patients with X chromosomes derived from the father or mother. These results suggest that X chromosomes from both parents may have less functional loss in growth and development, and the etiology may be more complex, so that single exogenous GH supplementation may afford only limited effects. Nevertheless, we did not undertake detailed sequencing and, so far, we are unable to implicate the mechanism underlying the influence of combined variation sources.

Another unexpected finding is that the mother's age and height at the time of the subject's birth might strongly influence body development and GH treatment. The clear mechanism is still to be explored. To our best knowledge, this is the first work showing the correlation between mother's age and height at the time of the patient's birth and GH treatment response, and no direct evidence was found for reference. The etiology of this finding can be explained as follow. First, the uterine size was found to be smaller if the child was born to an older mother, which can be expected. Consistently, the patient's prolactin level significantly decreased as the mother's age at birth increased. Prolactin ay partially impact the GH response to growth hormone-releasing hormone (*Losa et al., 1988*). Besides, elderly women tend to have chromosomal abnormality of oocyte, which leads to a high incidence rate of fetal chromosomal abnormality (*Ben-Meir et al., 2015*; *Xu et al., 2018*). However, it is still early to recommend for early childbirth towards mothers of potential TS children. The mother's height at the time of birth positively correlated with the GH therapy response, as indicated by the IGF-1 level after treatment, suggesting that patients with TS may benefit from a taller mother; however, the underlying mechanism needs to be explored. Intriguingly, the bone age difference at the baseline level negatively correlated with the mother's height at the time of birth, with no association observed following treatment. This may be attributed to the following reasons. Firstly, the *p*-value was not sufficiently small, and our sample size was limited; hence, this conclusion needs to be confirmed with more evidence. Secondly, the mother's increased height may suppress the bone development in the TS child, despite benefits for GH responses. Therefore, a negative correlation was observed at baseline, but the correlation was not observed after treatment. This hypothesis warrants further corroborative investigations.

The present study has some limitations. First, this study was conducted over an extremely brief timeframe (only 1 year), and the results may therefore not best predict the outcomes. Accordingly, we plan to perform an investigation with a more extended follow-up period. In addition, the age range of subjects varied greatly; this is to be expected, as mildly affected patients are diagnosed late, and presumably, those diagnosed late demonstrate poor growth potential. Their poor height outcome may be due to missed treatment. Our future investigation will accumulate more samples and narrow the age range. Moreover, based on the present findings, our ensuing investigation will focus on a

potentially effective strategy for those TS patients with retained X chromosomes derived from both parents, as well as high-risk children with TS sisters.

## CONCLUSION

In summary, the retained X-chromosome derived from both parents weakens the response to GH therapy. The mother's age and height at the time of birth can impact the patient's body/sexual development and GH treatment outcomes. Based on these data, considerable attention should be paid to analyzing the mother's age and height at the time of birth, as well as to the parental origin of the X-chromosome, prior to developing an effective treatment plan for TS.

## ABBREVIATIONS

| | |
|---|---|
| **TS** | Turner syndrome |
| **GH** | Growth hormone |
| **ALT** | Alanine aminotransferase |
| **AST** | Aspartate aminotransferase |
| **HDL** | High density lipoprotein |
| **LDL** | Low density lipoprotein |
| **TSH** | Thyroid-stimulating hormone |
| **LH** | Luteinizing hormone |
| **FSH** | Follicle-stimulating hormone |
| **HCG** | Human chorionic gonadotropin |
| **SHBG** | Sex hormone binding globulin |
| **X-STR** | X-chromosome short tandem repeats |
| **SD** | Standard deviation |

### Funding

This work was supported by The Fund of the Shanghai Municipal Commission of Health and Family Planning (No. 20174Y0007) and the Key and Special Project of Clinical Research and Cultivation of Shanghai Children's Hospital No. 2019YLYM07). Shanghai 2021 "Science and Technology Innovation Action Plan" medical innovation research special key project: No 21Y21901000 and "Three-Year Plan for Promoting Clinical Skills and Innovation in Municipal Hospitals of Shanghai", project for precise diagnosis and treatment of difficult diseases (No. SHDC2020CR2058B). The funders had no role in study design, data collection and analysis, decision to publish, or preparation of the manuscript.

### Grant Disclosures

The following grant information was disclosed by the authors:
Shanghai Municipal Commission of Health and Family Planning: 20174Y0007.
Key and Special Project of Clinical Research and Cultivation of Shanghai Children's Hospital: 2019YLYM07.

Science and Technology Innovation Action Plan: 21Y21901000.
Three-Year Plan for Promoting Clinical Skills and Innovation in Municipal Hospitals of Shanghai: SHDC2020CR2058B.

## Competing Interests

The authors declare that they have no competing interests.

## Author Contributions

- Ying Zhang analyzed the data, authored or reviewed drafts of the paper, and approved the final draft.
- Yongchen Yang performed the experiments, prepared figures and/or tables, and approved the final draft.
- Pin Li analyzed the data, authored or reviewed drafts of the paper, and approved the final draft.
- Sheng Guo conceived and designed the experiments, performed the experiments, prepared tables, and approved the final draft.

## Human Ethics

The following information was supplied relating to ethical approvals (*i.e.*, approving body and any reference numbers):

Children's Hospital of Shanghai/Shanghai Children's Hospital, Shanghai Jiao Tong University (Approval No: 2017R027-E04).

## Data Availability

The raw measurements are available in the Supplementary Files.

## Supplemental Information

Supplemental information for this article can be found online at http://dx.doi.org/10.7717/peerj.12354#supplemental-information.

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
