# Peer review of "Origin of the X-chromosome influences the development and treatment outcomes of Turner syndrome"

_PeerJ, doi:10.7717/peerj.12354_

## Round 0.1 · original submission · Major Revisions

We have received comments from three reviewers. The reviewers acknowledge the importance of this study. However, they have also found problems. I think their comments are reasonable and should be fixed in revision. Especially, statistic analysis is important for publication. In addition, the reviewers requested to improve the English expression. I request that these points be corrected.

Reviewer 1 ·

Basic reporting

Grammar and syntax needs some modifications.
Literature and citations are sufficient
Background could be improved- see Notes to Authors below. Evidence within the discussion (referring to other papers), context, and result reference ranges are lacking.
Figures and tables are easy to read, interpret.
Raw Data not available

Experimental design

Project is within "Aims and Scope" of the journal

The research question is well defined and certainly very meaningful, with great potential to fill a knowledge gap.

Methods should be provided in more detail, expanded - how were assays performed (were estradiol, LH via ICMA, what reference ranges were used and how did this correlate to age, puberty, sex. how were chromosomes of origin detected.

I think the investigation may not be as rigorous as anticipated.

Standard of ethics appear to have been followed.

Validity of the findings

Concern that the results are not correlated to what is known in this population. Research data/results are provided through Figures and Tables, but it is hard to determine if these data are truly statistically sound without reference ranges or standards provided.

Conclusions are concerning. Some sections of the discussion are inaccurate- see Notes to Authors below.

Conclusions are not supported by the results obtained in the study. Conclusions are based on extremely limited studies over a very brief timeframe (9 months) that may not best predict outcomes. It is unclear if the authors recognize this limitation, as it is not mentioned anywhere in the manuscript.

Additional comments

This is an interesting study by the authors on clinical effects of parental origin of the retained X chromosome(s) in patients with Turner Syndrome. The concept of using parent of origin studies is fascinating and there is likely a lot to be learned in this area. The authors are to be commended for their originality and thoughtful consideration of this project.

Strengths:
sizable number of study participants
thoughtful genetic evaluation for karyotype, parent of origin studies, extensive laboratory testing
Methods of TS diagnosis, treatment, and dose of therapy are clear

Additional details of this research project are unclear. On line 76, inclusion criteria reads, “suitable for GH” – please define this. It would also add value if you explain how parent of origin studies are conducted.

One concern I have is that the age range of subjects is quite wide. While this is expected, since mildly affected patients have a late diagnosis, and presumably those with late diagnosis also have less potential for growth. Their poorer height outcome may be due to missed treatment, not alternative factors. In addition, age of subject influences outcomes such as uterine size, growth rates, bone age progression; furthermore, the reference range in different ages is not the same, so cannot be compared.

Importantly, this manuscript does not include standards/reference ranges. For example, definitions for “small uterus” or “large uterus” are missing - these dimensions vary by age and pubertal status, at minimum. A 12 yo without spontaneous estrogen will be different than a 12 yo with spontaneous, endogenous estrogen. Specifically, this the range of estrogen levels was quite wide- which supports that some patients may have been on estrogen. This can impact the study results in multiple ways.

The authors mentioned a number of laboratory analyses that showed low levels. The manuscript would be further enriched if the authors explained why those reduced values are (or are not) clinically relevant. The authors also mention impact of retained maternal X chromosome on prolactin levels. Please provide background on why you are concerned about prolactin levels in patients with TS and how this impacts their management.

Some discussion points are unclear or poorly supported by evidence. Line 167- remove “Besides” and describe how this relationship is changed. Line 168- “GH treatment show restored IGF-1 and IGFBP-3 levels” – but baseline levels in TS are typically normal.

Importantly, the authors should comment if 9 months an adequate time frame to assess growth response in this population?

The manuscript would be greatly enhanced by also providing a section on further work to be done based on these results. Lastly, consider improvements in grammar, syntax, and English so that this paper appeals to a broad variety of English-speakers.

Reviewer 2 ·

Basic reporting

The English language throughout this manuscript could be improved upon by having a colleague who is proficient in English and familiar with the subject matter review your manuscript to insure the most appropriate use of wording and punctuation are addressed. Some examples where the language could be improved upon include lines 83,112,125,186,187,199,208 and more.

Your background and literature support would be stronger by providing more context of what has been demonstrated regarding height, specifically related to the SHOX gene and epigenetic reports, as well as parent of origin findings related to survivability as well as hypotheses of all survivors are mosaic from previously published literature.

Experimental design

Data regarding cases in which both parents of origin have X chromosome material present in the karyotype could be stronger by substantiating the parent of origin for the presence of the shox gene(s), and which del(X) parental origin material is present. By grouping the parent of origin and clarifying breakpoints in the tables, such as table 3, it would help the reader more clearly understand the role parent of origin is playing vs. genetic material deleted.

Validity of the findings

The conclusion would be stronger with more details of the substantiated clinical utility of these findings and how this research matters.

Reviewer 3 ·

Basic reporting

English language is awkward in many parts, sometimes making difficult to understand the meaning.
Too often you use very long periods! To give you an example, the periods between lines 136 and 140 or between 153 and 156 are excessively long. You must try to use shorter and concise periods, which will certainly be clearer and appreciated by the reader.
Moreover, there are many grammatical errors, even banal (e.g. commas, uppercase or lowercase, etc.).
Another example (line 147): "between two parents"???..Did you maybe mean "among parents"?
Author must submit their manuscript to a native English speaker to bring several important changes and to make their manuscript understandable.
You also didn’t even number the pages of the entire document, which would have facilitated the review process.

Experimental design

Writing of this paper is not the clearest.
Both introduction and discussion should focus on what this study adds to the existing literature.
There are some rough errors, such as the mismatch between the age range expressed at line 75 and line 122.
Another methodological error, in my opinion, is that of line 142: you have taken into consideration, in a fundamentally arbitrary way, only patients between the ages of 5.2 and 10 years. It would have been more appropriate to select sub-groups on the basis of the different growth rates expected in the different age groups.
Also, why did you choose just 9-month GH treatment as duration?
At line 131 it is assumed that patients had a low LH value. You can't make such a statement, as you consider both young and adult subjects, with only the latter with an overt ovarian damage, and therefore with a condition of hypergonadotropic hypogonadism!
At line 135 you claim to have divided patients into three groups, based on the retained X chromosome origin. What is the point, then, of having previously divided patients on the basis of different karyotype groups (Type 1-4)? You should clarify that.
Please add the normative values in Table 2.

Validity of the findings

The subject matter is certainly interesting. ot all conclusions, however, can be drawn with certainty! For example, at line 152, you write: "these results suggest that an elder pregnant woman...". It would be much more appropriate to use a conditional, such as could or may.."these results MAY or COULD suggest that an elder pregnant woman..."
Line 186: From "so far...to effects": Where is the verb in this sentence?
Then you draw conclusions about the link between the mother’s age and the lower prolactin level. Considering that a scientific paper has to be also clinically applicable, what do you find about low levels of prolactin in women? Do you know of any pathology related to hypoprolactinemia?!
In general, the conclusions you draw seem quite "forced", in order to try to add something new to the existing literature on the subject.

---

## Round 0.2 · Minor Revisions

Reviewers acknowledge improvements to the revised manuscript, but also find problems. I think the reviewers' comments further improve the paper.

Reviewer 2 ·

Basic reporting

No comment

Experimental design

no comment

Validity of the findings

Raw data: Detailed structural karyotype information is not provided, even in the raw data, esp in consideration of breakpoints in genes which may be disrupted.

Pg 15. Discussion regarding maternal age at the time of birth and height increase response should be substantiated further based on literature reports and possible underlying mechanisms to explain the etiology of this finding.

In addition, the maternal origin and paternal origin is not clarified in these cases of maternal age.

Pg 15. "Growth retardation tends to be more sever in patients with XrX,isoXq,and 45,X" yet the findings of this current study contradict these reports as isoXq would have X chromosomes from both parents- when in fact the Xp arm is truly only present from one parent, etc. It would be a stronger paper to substantiate regions from only one parent vs. chromosome origins from one or both.

Lastly, the conclusion of recommending early childbearing for mothers of children with potential TS is not sensible as most occurrences of TS (esp 45,X) are sporadic and not inherited.

Additional comments

no comment

---

## Round 0.3 · accepted · Accept

I am happy to inform you that your manuscript has been accepted.